# Decoding Decoders:
# Finding Optimal Representation Spaces
# For Unsupervised Similarity Tasks

## Abstract

Experimental evidence indicates that simple models outperform complex deep networks on many unsupervised similarity tasks. Introducing the concept of an *optimal representation space*, we provide a simple theoretical resolution to this apparent paradox. In addition, we present a straightforward procedure that, without any retraining or architectural modifications, allows deep recurrent models to perform equally well (and sometimes better) when compared to shallow models. To validate our analysis, we conduct a set of consistent empirical evaluations and introduce several new sentence embedding models in the process. Even though this work is presented within the context of natural language processing, the insights are readily applicable to other domains that rely on distributed representations for transfer tasks.

## 1 Introduction

Distributed representations have played a pivotal role in the current success of machine learning. In contrast with the symbolic representations of classical AI, distributed representation spaces can encode rich notions of semantic similarity in their distance measures, allowing systems to generalise to novel inputs. Methods to learn these representations have gained significant traction, in particular for modelling words (Mikolov et al., 2013). They have since been successfully applied to many other domains, including images (Girod et al., 2011; Razavian et al., 2014) and graphs (Kipf & Welling, 2016; Grover & Leskovec, 2016; Narayanan et al., 2017).

Using unlabelled data to learn effective representations is at the forefront of modern machine learning research. The Natural Language Processing (NLP) community in particular, has invested significant efforts in the construction (Mikolov et al., 2013; Pennington et al., 2014; Bojanowski et al., 2016; Joulin et al., 2017), evaluation (Baroni et al., 2014) and theoretical analysis (Levy & Goldberg, 2014) of distributed representations for words.

Recently, attention has shifted towards the unsupervised learning of representations for larger pieces of text, such as phrases (Yin & Schütze, 2015; Zhang et al., 2017), sentences (Kalchbrenner et al., 2014; Kiros et al., 2015; Tai et al., 2015; Hill et al., 2016; Arora et al., 2017), and entire paragraphs (Le & Mikolov, 2014). Some of this work simply sums or averages constituent word vectors to obtain a sentence representation (Mitchell & Lapata, 2010; Milajevs et al., 2014; Wieting et al., 2015; Arora et al., 2017), which is surprisingly effective but naturally cannot leverage any contextual information.

Another line of research has relied on a *sentence-level distributional hypothesis* (Polajnar et al., 2015), originally applied to words (Harris, 1954), which is an assumption that sentences which occur in similar contexts have a similar meaning. Such models often use an encoder-decoder architecture (Cho et al., 2014) to predict the adjacent sentences of any given sentence. Examples of such models include SkipThought (Kiros et al., 2015), which uses Recurrent Neural Networks (RNNs) for its encoder and decoders, and FastSent (Hill et al., 2016), which replaces the RNNs with simpler bag-of-words (BOW) versions.

Models trained in an unsupervised manner on large text corpora are usually applied to *supervised transfer tasks*, where the representation for a sentence forms the input to a supervised classification

problem, or to *unsupervised similarity tasks*, where the similarity (typically taken to be the cosine similarity) of two inputs is compared with corresponding human judgements of semantic similarity in order to inform some downstream process, such as information retrieval.

Interestingly, some researchers have observed that deep complex models like SkipThought tend to do well on supervised transfer tasks but relatively poorly on unsupervised similarity tasks, whereas for shallow log-linear models like FastSent the opposite is true (Hill et al., 2016; Conneau et al., 2017). It has been highlighted that this should be addressed by analysing the geometry of the representation space (Almahairi et al., 2015; Schnabel et al., 2015; Hill et al., 2016), however, to the best of our knowledge it has not been systematically attempted[1].

In this work we attempt to address the observed performance gap on unsupervised similarity tasks between representations produced by simple models and those produced by deep complex models. Our main contributions are as follows:

- We introduce the concept of an *optimal representation space*, in which the space has a similarity measure that is optimal with respect to the objective function.

- We show that models with log-linear decoders are usually evaluated in their optimal space, while recurrent models are not. This effectively explains the performance gap on unsupervised similarity tasks.

- We show that, when evaluated in their optimal space, recurrent models close that gap. We also provide a procedure for extracting this optimal space using the decoder hidden states.

- We validate our findings with a series of consistent empirical evaluations utilising a single publicly available codebase.

## 2 OPTIMAL REPRESENTATION SPACE

We begin by considering a general problem of learning a conditional probability distribution $P_{\text{model}}(y \mid x)$ over the output symbols $y \in \mathcal{Y}$ given the input symbols $x \in \mathcal{X}$.

**Definition 1.** A space $\mathcal{H}$ combined with a similarity measure $\rho : \mathcal{H} \times \mathcal{H} \mapsto \mathbb{R}$ in which semantically close symbols $s_i, s_j \in \mathcal{S}$ have representations $\mathbf{h}_i, \mathbf{h}_j \in \mathcal{H}$ that are close in $\rho$ is called a *distributed representation space* (Goodfellow et al., 2016).

In general, a distributed representation of a symbol $s$ is obtained via some function $\mathbf{h}_s = f(s; \theta_f)$, parametrised by weights $\theta_f$. Distributed representations of the input symbols are typically found as the layer activations of a Deep Neural Network (DNN). One can imagine running all possible $x \in \mathcal{X}$ through a DNN and using the activations $\mathbf{h}_x$ of the $n^{th}$ layer as vectors in $\mathcal{H}_x$:

$$\mathcal{H}_x = \left\{ \mathbf{h}_x = \text{Activation}^{(n)}(x) \mid x \in \mathcal{X} \right\}.$$

The distributed representation space of the output symbols $\mathcal{H}_y$ can be obtained via some function $\mathbf{h}_y = g(y; \theta_g)$ that does not depend on the input symbol $x$, e.g. a row of the softmax projection matrix that corresponds to the output $y$.

In practice, although $\mathcal{H}$ obtained in such a manner with a reasonable vector similarity $\rho$ (such as cosine or Euclidean distance) forms a distributed representation space, there is no *a priori* reason why an arbitrary choice of a similarity function would be appropriate given $\mathcal{H}$ and the model's objective. There is no analytic guarantee, for arbitrarily chosen $\mathcal{H}$ and $\rho$, that small changes in semantic similarity of symbols correspond to small changes in similarity $\rho$ between their vector representations in $\mathcal{H}$ and vice versa. This motivates Definition 2.

**Definition 2.** A space $\mathcal{H}$ equipped with a similarity measure $\rho$ such that $\log P_{\text{model}}(y \mid x) \propto \rho(\mathbf{h}_y, \mathbf{h}_x)$ is called an *optimal representation space*.

---

[1] Although the models in Wieting & Gimpel (2017) are trained on a supervised dataset of aligned sentences or paraphrases (and are therefore distinct from our unsupervised setting), it is interesting to note that they observe a similar discrepancy between the performance of RNNs and simply averaging word vectors.

In words, if a model has an optimal representation space, the conditional log-probability of an output symbol $y$ given an input symbol $x$ is proportional to the similarity $\rho(\mathbf{h}_y, \mathbf{h}_x)$ between their corresponding vector representations $\mathbf{h}_y, \mathbf{h}_x \in \mathcal{H}$.

For example, consider the following standard classification model

$$P_{\text{model}}(y \mid x) = \frac{\exp\left(\mathbf{u}_y \cdot \text{DNN}(x)\right)}{\sum_{y'} \exp\left(\mathbf{u}_{y'} \cdot \text{DNN}(x)\right)} \tag{1}$$

where $\mathbf{u}_y$ is the $y^{th}$ row of the output projection matrix $\mathbf{U}$.

If $\mathcal{H}_x = \{\text{DNN}(x) \mid x \in \mathcal{X}\}$ and $\mathcal{H}_y = \{\mathbf{u}_y \mid y \in \mathcal{Y}\}$, then $\mathcal{H} = \mathcal{H}_x \cup \mathcal{H}_y$ equipped with $\rho(\mathbf{h}_1, \mathbf{h}_2) = \mathbf{h}_1 \cdot \mathbf{h}_2$ (the dot product) is an optimal representation space. Note that if the exponents of Equation (1) contained Euclidean distance, then we would find $\log P_{\text{model}}(y \mid x) \propto ||\mathbf{u}_y - \text{DNN}(x)||_2$. The optimal representation space would then be equipped with Euclidean distance as its optimal distance measure $\rho$. This easily extends to any other distance measures desired to be induced on the optimal representation space.

Let us elaborate on why Definition 2 is a reasonable definition of an optimal space. Let $x_1, x_2 \in \mathcal{X}$ be the input symbols and $y_1, y_2 \in \mathcal{Y}$ their corresponding outputs. Using

$$\mathbf{a} \overset{\rho}{\sim} \mathbf{b}$$

to denote that $\mathbf{a}$ and $\mathbf{b}$ are close under $\rho$, a reasonable model trained on a subset of $(\mathcal{X}, \mathcal{Y})$ will ensure that $\mathbf{h}_{x_1} \overset{\rho}{\sim} \mathbf{h}_{y_1}$ and $\mathbf{h}_{x_2} \overset{\rho}{\sim} \mathbf{h}_{y_2}$. If $x_1$ and $x_2$ are semantically close and assuming semantically close input symbols have similar outputs, we also have that $\mathbf{h}_{x_1} \overset{\rho}{\sim} \mathbf{h}_{y_2}$ and $\mathbf{h}_{x_2} \overset{\rho}{\sim} \mathbf{h}_{y_1}$. Therefore it follows that $\mathbf{h}_{x_1} \overset{\rho}{\sim} \mathbf{h}_{x_2}$ (and $\mathbf{h}_{y_1} \overset{\rho}{\sim} \mathbf{h}_{y_2}$). Putting it differently, semantic similarity of input and output symbols translates into closeness of their distributed representations under $\rho$, in a way that is consistent with the model.

Note that any model $P_{\text{model}}(y \mid x)$ parametrised by a continuous function can be approximated by a function in the form of Equation (1). It follows that any model that produces a probability distribution has an optimal representation space. Also note that the optimal space for the inputs does not necessarily have to come from the final layer before the softmax projection but instead can be constructed from any layer, as we now demonstrate.

Let $n$ be the index of the final activation before the softmax projection and let $k \in \{1, \ldots, n\}$. We split the network into three parts:

$$\text{softmax}\left(\mathbf{U}F_n\left(G_k(x)\right)\right) \tag{2}$$

where $G_k$ contains first $k$ layers, $F_n$ contains the remaining $n - k$ layers and $\mathbf{U}$ is the softmax projection matrix. Let the space for inputs $\mathcal{H}_x$ be defined as

$$\mathcal{H}_x = \{G_k(x) \mid x \in \mathcal{X}\}$$

and the space for outputs $\mathcal{H}_y$ defined as

$$\mathcal{H}_y = \{\mathbf{u}_y \mid y \in \mathcal{Y}\}.$$

Their union $\mathcal{H} = \mathcal{H}_x \cup \mathcal{H}_y$ equipped with $\rho(\mathbf{h}_1, \mathbf{h}_2) = J(\mathbf{h}_1) \cdot J(\mathbf{h}_2)$ where

$$J(\mathbf{h}) = \begin{cases} F_n(\mathbf{h}) & \text{if } \mathbf{h} \in \mathcal{H}_x \\ \mathbf{h} & \text{otherwise} \end{cases}$$

is again an optimal representation space. We will show a specific example where this holds in Section 3.3.

# 3 Optimal Spaces for Sentence Representations

For the remainder of this paper, we focus on unsupervised models for learning distributed representations of sentences, an area of particular interest in NLP.

### 3.1 Background

Let $S = (s_1, s_2, \ldots, s_N)$ be a corpus of contiguous sentences where each sentence $s_i = w_{s_i}^1 w_{s_i}^2 \ldots w_{s_i}^{\tau_{s_i}}$ consists of words from a pre-defined vocabulary $V$ of size $|V|$.

We transform the corpus into a set of pairs $D = \{(s_i, c_i)\}_{i=1}^N$, where $s_i \in S$ and $c_i$ is a context of $s_i$. The context usually (but not necessarily) contains some number of surrounding sentences of $s_i$, e.g. $c_i = (s_{i-1}, s_{i+1})$.

We are interested in modelling the probability of a context $c$ given a sentence $s$. In general

$$P_{\text{model}}(c \mid s; \theta) = \prod_{t=1}^{\tau_c} P_{\text{model}}(w_c^t \mid w_c^{t-1}, \ldots, w_c^1, s; \theta). \tag{3}$$

One popular way to model $P(c \mid s)$ for sentence-level data is suggested by the encoder-decoder framework. The encoder $\mathcal{E}$ produces a fixed-length vector representation $\mathbf{h}_s^{\mathcal{E}} = \mathcal{E}(s)$ for a sentence $s$ and the decoder gives a context prediction $\hat{c} = \mathcal{D}(\mathbf{h}_s^{\mathcal{E}})$ from that representation.

Due to a clear architectural separation between $\mathcal{E}$ and $\mathcal{D}$, it is common to take $\mathbf{h}_s^{\mathcal{E}}$ as a representation of a sentence $s$ in the downstream tasks. Furthermore, since $\mathbf{h}_s^{\mathcal{E}}$ is usually encoded as a vector, such representations are often compared via simple similarity measures, such as dot product or cosine similarity.

### 3.2 Log-Linear Decoders

We first consider encoder-decoder architectures with a log-linear BOW decoder for the context. Let $\mathbf{h}_i = \mathcal{E}(s_i)$ be a sentence representation of $s_i$ produced by some encoder $\mathcal{E}$. The nature of $\mathcal{E}$ is not important for our analysis; for concreteness, the reader can consider a model such as FastSent (Hill et al., 2016), where $\mathcal{E}$ is a BOW (sum) encoder.

In the case of the log-linear BOW decoder, words are conditionally independent of the previously occurring sequence, thus Equation (3) becomes

$$P_{\text{model}}(c_i|s_i; \theta) = \prod_{w \in c_i} P_{\text{model}}(w|s_i; \theta) = \prod_{w \in c_i} \frac{\exp\left(\mathbf{u}_w \cdot \mathbf{h}_i\right)}{\sum_{w' \in V} \exp\left(\mathbf{u}_{w'} \cdot \mathbf{h}_i\right)} = \frac{\prod_{w \in c_i} \exp\left(\mathbf{u}_w \cdot \mathbf{h}_i\right)}{|c_i| \sum_{w' \in V} \exp\left(\mathbf{u}_{w'} \cdot \mathbf{h}_i\right)}. \tag{4}$$

where $\mathbf{u}_w \in \mathbb{R}^d$ is the output word embedding for a word $w$ and $\mathbf{h}_i$ is the encoder output. (Biases are omitted for brevity.)

The objective is to maximise the model probability of contexts $c_i$ given sentences $s_i$ across the corpus $D$, which corresponds to finding the Maximum Likelihood Estimator (MLE) for the trainable parameters $\theta$:

$$\theta_{\text{MLE}} = \arg\max_{\theta} \prod_{(s_i, c_i) \in D} P_{\text{model}}(c_i|s_i; \theta). \tag{5}$$

By switching to the negative log-likelihood and inserting the above expression, we arrive at the following optimisation problem:

$$\theta_{\text{MLE}} = \arg\min_{\theta} \left[ - \sum_{(s_i, c_i) \in D} \left( \sum_{w \in c_i} \mathbf{u}_w \cdot \mathbf{h}_i + |c_i| \log \sum_{w' \in V} \exp\left(\mathbf{u}_{w'} \cdot \mathbf{h}_i\right) \right) \right]. \tag{6}$$

Noticing that

$$\sum_{w \in c_i} \mathbf{u}_w \cdot \mathbf{h}_i = \left( \sum_{w \in c_i} \mathbf{u}_w \right) \cdot \mathbf{h}_i = \mathbf{c}_i \cdot \mathbf{h}_i, \tag{7}$$

we see that the objective in Equation (6) forces the sentence representation $\mathbf{h}_i$ to be similar under dot product to its context representation $\mathbf{c}_i$, which is simply the sum of the output embeddings of the context words. Simultaneously, output embeddings of words that do not appear in the context of a sentence are forced to be dissimilar to its representation.

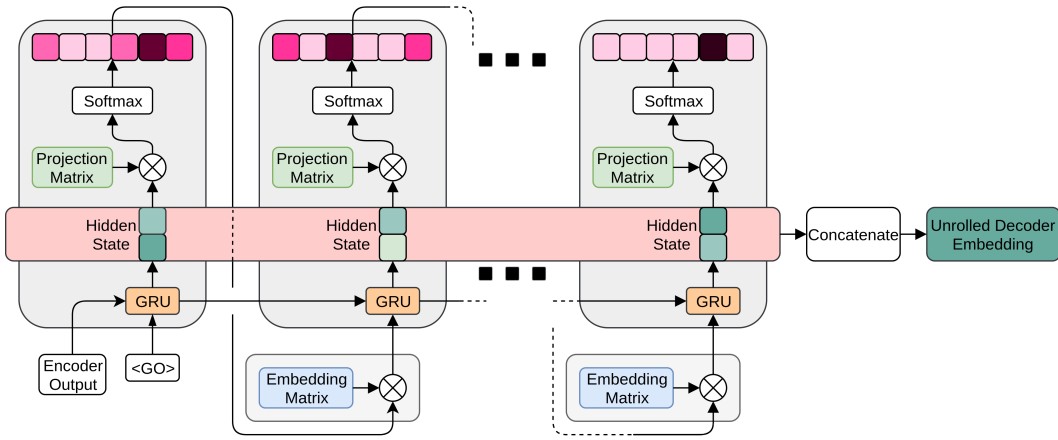

Figure 1: Unrolling a RNN decoder at inference time. The initial hidden state for the decoder is typically the encoder output, either the recurrent cell final state for a RNN encoder, or the sum of the input word embeddings for a BOW encoder. At the first time step, a learned `<GO>` token is presented as the input. In subsequent time steps, a probability-weighted sum over word vectors is used. The decoder is then unrolled for a fixed number of steps. The hidden states are then concatenated to produce the unrolled decoder embedding. In the models evaluated in Section 4, this process is performed for the RNN corresponding to the previous and next sentences. The sentence representation is then taken as the concatenation across both RNNs.

Using $\overset{\text{dot}}{\sim}$ to denote *close under dot product*, we find that if two sentences $s_i$ and $s_j$ have similar contexts, then $\mathbf{h}_i \overset{\text{dot}}{\sim} \mathbf{c}_j$ and $\mathbf{h}_j \overset{\text{dot}}{\sim} \mathbf{c}_i$. The objective function in Equation (6) ensures that $\mathbf{h}_i \overset{\text{dot}}{\sim} \mathbf{c}_i$ and $\mathbf{h}_j \overset{\text{dot}}{\sim} \mathbf{c}_j$. Therefore, it follows that $\mathbf{h}_i \overset{\text{dot}}{\sim} \mathbf{h}_j$.

Putting it differently, sentences that occur in related contexts are assigned representations that are similar under the dot product. Hence we see that the encoder output equipped with the dot product constitutes an optimal representation space as defined in Section 2.

### 3.3 RECURRENT SEQUENCE DECODERS

Another common choice for the context decoder is an RNN decoder

$$\mathbf{h}^t = \text{RNNCell}\left(\mathbf{v}^t, \mathbf{h}^{t-1}\right), \quad \mathbf{h}^0 = \mathbf{h}_i \tag{8}$$

where $\mathbf{h}_i = \mathcal{E}(s_i)$ is the encoder output. The specific structure of $\mathcal{E}$ is again not important for our analysis. (When $\mathcal{E}$ is also an RNN, this is similar to SkipThought (Kiros et al., 2015).)

The time unrolled states of decoder are converted to probability distributions over the vocabulary, conditional on the sentence $s_i$ and all the previously occurring words. Equation (3) becomes

$$P_{\text{model}}(c_i|s_i;\theta) = \prod_{t=1}^{\tau_{c_i}} P_{\text{model}}(w^t|w^{t-1},\ldots,w^1,s_i;\theta) = \prod_{t=1}^{\tau_{c_i}} \frac{\exp\left(\mathbf{u}_{w^t}\cdot\mathbf{h}^t\right)}{\sum_{w'\in V}\exp\left(\mathbf{u}_{w'}\cdot\mathbf{h}^t\right)} \tag{9}$$

Similarly to Equation (6), MLE for the model parameters $\theta$ can be found as

$$\theta_{\text{MLE}} = \arg\min_{\theta}\left[-\sum_{(s_i,c_i)\in D}\sum_{t=1}^{\tau_{c_i}}\left(\mathbf{u}_{w^t}\cdot\mathbf{h}^t + \log\sum_{w'\in V}\exp\left(\mathbf{u}_{w'}\cdot\mathbf{h}^t\right)\right)\right]. \tag{10}$$

Using $\oplus$ to denote *vector concatenation*, we note that

$$\sum_{t=1}^{\tau_{c_i}}\mathbf{u}_{w^t}\cdot\mathbf{h}^t = \left(\bigoplus_{t=1}^{\tau_{c_i}}\mathbf{u}_{w^t}\right)\cdot\left(\bigoplus_{t=1}^{\tau_{c_i}}\mathbf{h}^t\right) = \mathbf{c}_i\cdot\mathbf{h}_i^{\mathcal{D}}, \tag{11}$$

where the sentence representation $\mathbf{h}_i^{\mathcal{D}}$ is now an *ordered* concatenation of the hidden states of the decoder and the context representation $\mathbf{c}_i$ is an *ordered* concatenation of the output embeddings of

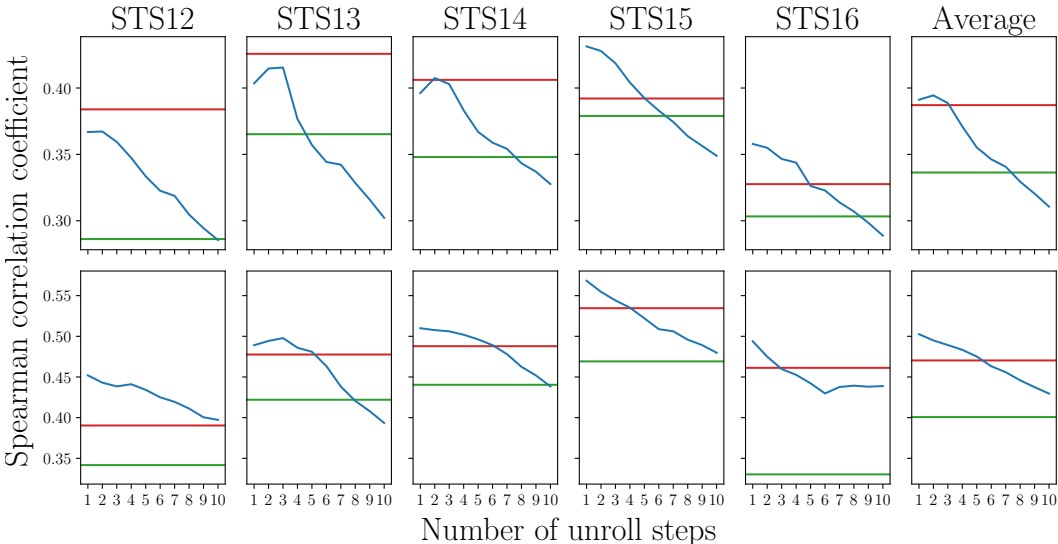

Figure 2: Performance on the STS tasks depending on the number of unrolled hidden states of the decoders, using dot product as the similarity measure. The top row presents results for the RNN encoder and the bottom row for the BOW encoder. **Red:** Raw encoder output with BOW decoder. **Green:** Raw encoder output with RNN decoder. **Blue:** Unrolled RNN decoder output. Independent of the encoder architecture, unrolling even a single state of the decoder always outperforms the raw encoder output with RNN decoder, and almost always outperforms the raw encoder output with BOW decoder for some number of unrolls.

the context words. Hence we can come to the same conclusion as in the log-linear case, except we have order-sensitive representations as opposed to unordered ones. As before, $\mathbf{h}_i^{\mathcal{D}}$ is forced to be similar to the context $\mathbf{c}_i$ under dot product, and is made dissimilar to sequences of $\mathbf{u}_{w'}$ that do not appear in the context.

The "transitivity" argument from Section 3.2 remains intact, except the length of decoder hidden state sequences might differ from sentence to sentence. To avoid this problem, we can formally treat them as infinite-dimensional vectors in $\ell^2$ with only a finite number of initial components occupied by the sequence and the rest set to zero. Alternatively, we can agree on the maximum sequence length, which in practice can be determined from the training corpus.

Regardless, the above space of unrolled concatenated decoder states, equipped with dot product, is the optimal representation space for models with recurrent decoders. Consequently, this space could be a much better candidate for unsupervised similarity tasks.

We refer to the method of accessing the decoder states at every time step as *unrolling the decoder*, illustrated in Figure 1. Note that accessing the decoder output does not require re-architecting or retraining the model, yet gives a potential performance boost on unsupervised similarity tasks almost for free. We will demonstrate the effectiveness of this technique empirically in Section 5.

## 4    EXPERIMENTAL SETUP

We have seen in Section 2 that the optimal representation space for a given model depends on the choice of decoder architecture. To support this theory, we train several encoder-decoder architectures for sentences with the decoder types analysed in Section 3, and evaluate them on downstream tasks using both their optimal space and the standard space of the encoder output as the sentence representations.

**Models and training.**   Each model has an encoder for the current sentence, and decoders for the previous and next sentences. As our analysis is independent of encoder type, we train and evaluate models with BOW and RNN encoders, two common choices in the literature for sentence representation learners (Hill et al., 2016; Kiros et al., 2015). The BOW encoder is the sum of word vectors (Hill et al., 2016). The RNN encoder and decoders are Gated Recurrent Units (GRUs) (Cho et al., 2014).

Table 1: Performance of different architectures and sentence representations on unsupervised similarity tasks using dot product as the similarity measure. On each task, the highest performing setup for each encoder type is highlighted in bold and the highest performing setup overall is underlined. All reported values indicate Pearson/Spearman correlation coefficients for the task. **RNN encoder:** Unrolling the RNN decoders using the concatenation of the decoder hidden states (RNN-concat) dramatically improves the performance across all tasks compared to using the raw encoder output (RNN-RNN), validating the theoretical justification presented in Section 3.3. **BOW encoder:** Unrolling the RNN decoders improves performance overall, however, the improvement is less drastic than that observed for the RNN encoder, which we discuss further in the main text.

| Encoder | Decoder | STS12 | STS13 | STS14 | STS15 | STS16 |
|---------|---------|-------|-------|-------|-------|-------|
| | BOW | 0.286/**0.384** | 0.381/**0.426** | 0.365/**0.406** | 0.262/0.392 | 0.260/0.328 |
| RNN | RNN | 0.267/0.286 | 0.371/0.365 | 0.357/0.348 | 0.379/0.379 | 0.313/0.303 |
| | RNN-concat | **0.335**/0.359 | **0.411**/0.415 | **0.413**/0.403 | **0.414**/**0.419** | **0.326**/**0.347** |
| | BOW | 0.351/0.390 | 0.418/0.478 | 0.442/0.488 | 0.455/0.535 | 0.370/**0.461** |
| BOW | RNN | 0.310/0.342 | 0.365/0.422 | 0.396/0.440 | 0.412/0.469 | 0.281/0.330 |
| | RNN-concat | **0.422**/**0.438** | **0.478**/**0.498** | **0.498**/**0.506** | **0.512**/**0.544** | **0.402**/0.460 |

Using the notation ENC-DEC, we train RNN-RNN, RNN-BOW, BOW-BOW, and BOW-RNN models. For each encoder-decoder combination, we test several methods of extracting sentence representations to be used in the downstream tasks. First, we use the standard choice of the final output of the encoder as the sentence representation. In addition, for models that have RNN decoders, we unroll between 1 and 10 decoder hidden states. Specifically, when we unroll $n$ decoder hidden states, we take the first $n$ hidden states from each of the decoders and concatenate them in order to get the resulting sentence representation. We refer to these representations as *-RNN-concat.

All models are trained on the Toronto Books Corpus (Zhu et al., 2015), a dataset of 70 million ordered sentences from over 7,000 books. The sentences are pre-processed such that tokens are lower case and splittable by space.

**Evaluation tasks.** We use the SentEval tool (Conneau et al., 2017) to benchmark sentence embeddings on both supervised and unsupervised transfer tasks. The supervised tasks in SentEval include paraphrase identification (MSRP) (Dolan et al., 2004), movie review sentiment (MR) (Pang & Lee, 2005), product review sentiment (CR), (Hu & Liu, 2004)), subjectivity (SUBJ) (Pang & Lee, 2004), opinion polarity (MPQA) (Wiebe et al., 2005), and question type (TREC) (Voorhees, 2002; Roth & Li, 2003). In addition, there are two supervised tasks on the SICK dataset, entailment and relatedness (denoted SICK-E and SICK-R) (Marelli et al., 2014). For the supervised tasks, SentEval trains a logistic regression model with 10-fold cross-validation using the model's embeddings as features.

The unsupervised Semantic Textual Similarity (STS) tasks are STS12-16 (Cer et al., 2017; Agirre et al., 2012; 2013; 2014; Agirre, 2015; Agirre et al., 2016), which are scored in the same way as SICK-R but without training a new supervised model; in other words, the embeddings are used to directly compute similarity. We use dot product to compute similarity as indicated by our analysis; results and discussion using cosine similarity, which is canonical in the literature, are presented in Appendix B. For more details on all tasks and the evaluation strategy, see Conneau et al. (2017).

**Implementation and hyperparameters.** Our goal is to study how different decoder types affect the performance of sentence embeddings on various tasks. To this end, we use identical hyperparameters and architecture for each model (except encoder and decoder types), allowing for a fair head-to-head comparison. Specifically, for RNN encoders and decoders we use a single layer GRU with layer normalisation (Ba et al., 2016). All the weights (including word embeddings) are initialised uniformly over $[-0.1, 0.1]$ and trained with Adam without weight decay or dropout (Kingma & Ba, 2014). Sentence length is clipped or zero-padded to 30 tokens and end-of-sentence tokens are used throughout training and evaluation. Following Kiros et al. (2015), we use a vocabulary size of $20k$ with vocabulary expansion, 620-dimensional word embeddings, and 2400 hidden units in all RNNs.

## 5 RESULTS

Performance of the unrolled models on the STS tasks is presented in Figure 2. We note that unrolling even a single state of the decoder always improves the performance over the raw encoder output with the RNN decoder, and nearly always does so for the BOW decoder for some number of unrolled hidden states.

Table 2: Performance of different architectures and sentence representations on supervised transfer tasks. On each task, the highest performing setup for each encoder type is highlighted in bold and the highest performing setup overall is underlined. All reported values indicate test accuracy on the task, except for SICK-R where we report the Pearson correlation with human-provided scores. Note that the analysis in Section 3 is not readily applicable here, as instead of using a similarity measure in the representation space directly, the supervised transfer tasks train an entirely new model on top the chosen representation.

| Encoder | Decoder | MR | CR | MPQA | SUBJ | SST | TREC | MRPC | SICK-R | SICK-E |
|---------|---------|-------|-------|-------|-------|-------|-------|-------|--------|--------|
| | BOW | 75.78 | 79.34 | 86.25 | 90.77 | 81.99 | 84.60 | 70.55 | 0.80 | 78.81 |
| RNN | RNN | **77.06** | 81.77 | **88.59** | **92.56** | **82.65** | 86.60 | **71.94** | **0.83** | **81.10** |
| | RNN-concat | 76.20 | **82.07** | 85.96 | 91.80 | 80.83 | **87.20** | 71.59 | 0.82 | 80.35 |
| | BOW | 76.16 | 81.14 | 87.03 | 92.77 | 81.66 | 84.20 | 71.07 | 0.84 | 80.58 |
| BOW | RNN | 76.05 | **82.07** | 85.80 | 92.13 | 80.83 | 87.20 | 72.99 | 0.82 | 78.87 |
| | RNN-concat | **77.27** | 82.04 | **88.74** | **92.88** | 81.82 | **89.60** | **73.68** | **0.85** | **82.26** |

We observe that the performance tends to peak around 2-3 hidden states and fall off afterwards. In principle, one might expect the peak to be around the average sentence length of the corpus. A possible explanation of this behaviour is the "softmax drifting effect". As there is no context available at inference time, we generate the word embedding for the next time step using the softmax output from the previous time step (see Figure 1). Given that for any sentence, there is no single correct context, the probability distribution over the next words in that context will be multi-modal. This will flatten the softmax and produce inputs for the decoder that diverge from the inputs it expects (i.e. word vectors for the vocabulary). Further work is needed to understand this and other possible causes in detail.

Performance across unsupervised similarity tasks is presented in Table 1 and performance across supervised transfer tasks is presented in Table 2. For the unrolled architectures, in these tables we report on the one that performs best on the STS tasks. When the encoder is an RNN, the supervised transfer results validate our claims in Section 3.3. The results are less conclusive when the encoder is a BOW. We believe this is caused by the simplicity of the BOW encoder forcing its outputs to obey the sentence-level distributional hypothesis irrespective of decoder type, resulting in multiple candidates for the optimal representation space, but this should be investigated with a detailed analysis in future work.

In addition, see Appendix A for a comparison with the original SkipThought results from the literature, and Appendix B for results using cosine similarity rather than dot product as the similarity measure in STS tasks, as is the canonical choice.

When we look at the performance on supervised transfer in Table 2, combined with the similarity results in Table 1, we see that the notion that models cannot be good at both supervised transfer and unsupervised similarity tasks needs refining; for example, RNN-RNN achieves strong performance on supervised transfer, while RNN-RNN-concat achieves strong performance on unsupervised similarity. In general, our results indicate that a single model may be able to perform well on different downstream tasks, provided that the representation spaces chosen for each task are allowed to differ.

Curiously, the unusual combination of a BOW encoder and concatenation of the RNN decoders leads to the best performance on most benchmarks, even slightly exceeding that of some supervised models on some tasks (Conneau et al., 2017). This architecture may be worth investigating.

## 6 CONCLUSION

In this work, we introduced the concept of an optimal representation space, where semantic similarity directly corresponds to distance in that space, in order to shed light on the performance gap between simple and complex architectures on downstream tasks. In particular, we studied the space of initial hidden states to BOW and RNN decoders (typically the outputs of some encoder) and how that space relates to the training objective of the model.

For BOW decoders, the optimal representation space is precisely the initial hidden state of the decoder equipped with dot product, whereas for RNN decoders it is not. Noting that it is precisely these spaces that have been used for BOW and RNN decoders has led us to a simple explanation for the observed performance gap between these architectures, namely that the former has been evaluated in its optimal representation space, whereas the latter has not.

Furthermore, we showed that any neural network that outputs a probability distribution has an optimal representation space. Since a RNN does produce a probability distribution, we analysed its objective function which motivated a procedure of unrolling the decoder. This simple method allowed us to extract representations that are provably optimal under dot product, without needing to retrain the model.

We then validated our claims by comparing the empirical performance of different architectures across transfer tasks. In general, we observed that unrolling even a single state of the decoder always outperforms the raw encoder output with RNN decoder, and almost always outperforms the raw encoder output with BOW decoder for some number of unrolls. This indicates different vector embeddings can be used for different downstream tasks depending on what type of representation space is most suitable, potentially yielding high performance on a variety of tasks from a single trained model.

Although our analysis of decoder architectures was restricted to BOW and RNN, others such as convolutional (Xu et al., 2016) and graph (Kipf & Welling, 2016) decoders are more appropriate for many tasks. Similarly, although we focus on Euclidean vector spaces, hyperbolic vector spaces (Nickel & Kiela, 2017), complex-valued vector spaces (Trouillon et al., 2016) and spinor spaces (Kanjamapornkul et al., 2017) all have beneficial modelling properties. In each case, although an optimal representation space should exist, it is not clear if the intuitive space and similarity measure is the optimal one. However, there should at least exist a mapping from the intuitive choice of space to the optimal space using a transformation provided by the network itself, as we showed with the RNN decoder. Evaluating in this space should further improve performance of these models. We leave this for future work.

Ultimately, a good representation is one that makes a subsequent learning task easier. For unsupervised similarity tasks, this essentially reduces to how well the model separates objects in the chosen representation space, and how appropriately the similarity measure compares objects in that space. Our findings lead us to the following practical advice: i) Use a simple model architecture where the optimal representation space is clear by construction, or ii) use an arbitrarily complex model architecture and analyse the objective function to reveal, for a chosen vector representation, an appropriate similarity metric.

We hope that future work will utilise a careful understanding of what similarity means and how it is linked to the objective function, and that our analysis can be applied to help boost the performance of other complex models.

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

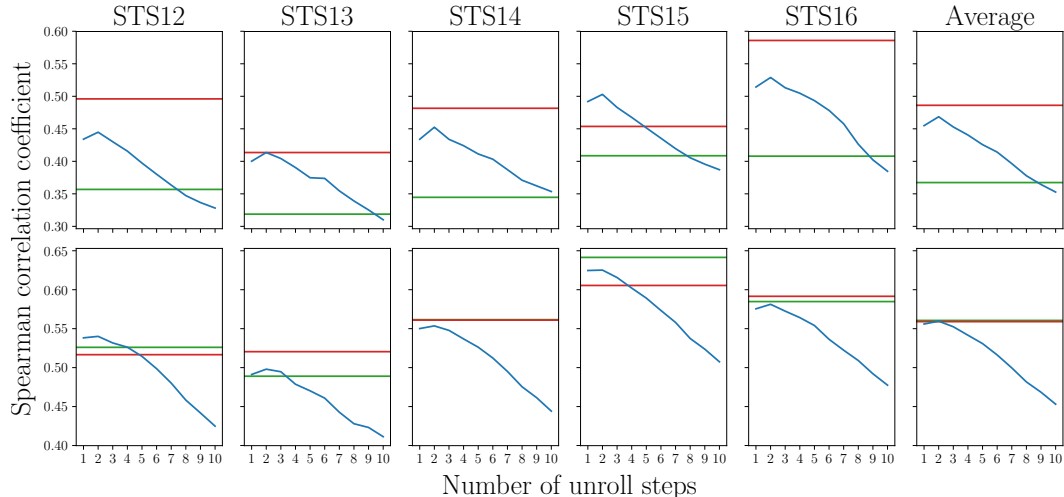

Figure 3: Performance on the STS tasks depending on the number of unrolled hidden states of the decoders, using cosine similarity as the similarity measure. The top row presents results for the RNN encoder and the bottom row for the BOW encoder. **Red:** Raw encoder output with BOW decoder. **Green:** Raw encoder output with RNN decoder. **Blue:** Unrolled RNN decoder output. For both RNN and BOW encoders, unrolling the decoder strictly outperforms *-RNN for almost every number of unroll steps, and perform nearly as well as or better than *-BOW.

## A    COMPARISON WITH SKIPTHOUGHT

Table 3: Performance of the SkipThought model, with and without layer normalisation (Kiros et al., 2015; Ba et al., 2016), compared against the RNN-RNN model used in our experimental setup. On each task, the highest performing model is highlighted in bold. For SICK-R, we report the Pearson correlation, and for STS14 we report the Pearson/Spearman correlation with human-provided scores. For all other tasks, reported values indicate test accuracy. † indicates results taken from Conneau et al. (2017). ‡ indicates our results from running SentEval on the model downloaded from Ba et al. (2016)'s publicly available codebase (https://github.com/ryankiros/layer-norm). We attribute the discrepancies in performance to differences in experimental setup or implementation. However, we expect our unrolling procedure to also boost SkipThought's performance on unsupervised similarity tasks, as we show for RNN-RNN in our fair single-codebase comparisons in the main text.

| Model | MR | CR | SUBJ | MPQA | SST | TREC | MRPC | SICK-R | SICK-E | STS14 |
|---|---|---|---|---|---|---|---|---|---|---|
| SkipThought † | 76.5 | 80.1 | 93.6 | 87.1 | 82.0 | **92.2** | 73.0 | **0.86** | **82.3** | 0.29/0.35 |
| SkipThought-LN † | **79.4** | **83.1** | **93.7** | **89.3** | 82.9 | 88.4 | - | **0.86** | 79.5 | **0.44/0.45** |
| SkipThought-LN ‡ | 78.6 | 82.2 | 92.9 | 89.1 | **83.8** | 87.0 | **73.2** | **0.86** | 81.2 | 0.41/0.40 |
| RNN-RNN | 77.1 | 81.8 | 92.6 | 88.6 | 82.7 | 86.6 | 71.9 | 0.83 | 81.1 | 0.35/0.35 |

## B    COSINE SIMILARITY ON STS TASKS

As discussed in Section 3, the objective function is maximising the dot product between the BOW decoder/unrolled RNN-decoder and the context. However, as other researchers in the field and the STS tasks specifically use cosine similarity by default, we present the results using cosine similarity in Table 4 and the results for different numbers of unrolled hidden decoder states in Figure 3.

Although the results in Table 4 are consistent with the dot product results in Table 1, the overall performance across STS tasks is noticeably lower when dot product is used instead of cosine similarity to determine semantic similarity. Switching from using cosine similarity to dot product transitions from considering only angle between two vectors, to also considering their length. Empirical studies have indicated that the length of a word vector corresponds to how sure of its context the model that produces it is. This is related to how often the model has seen the word, and how many different contexts it appears in (for example, the word vectors for "January" and "February" have similar norms, however, the word vector for "May" is noticeably smaller) (Schakel & Wilson, 2015).

Table 4: Performance of different architectures and sentence representations on unsupervised similarity tasks using the cosine similarity between two vectors as the measure of their similarity. On each task, the highest performing setup for each encoder type is highlighted in bold and the highest performing setup overall is underlined. All reported values indicate Pearson/Spearman correlation coefficients for the task. **RNN encoder:** Using the raw encoder output (RNN-RNN) achieves the lowest performance across all tasks. Unrolling the RNN decoders dramatically improves the performance across all tasks compared to using the raw encoder RNN output, validating the theoretical justification presented in Section 3.3. **BOW encoder:** We do not observe the same uplift in performance from unrolling the RNN encoder compared to the encoder output. This is consistent with our findings when using dot product (see Table 1).

| Encoder | Decoder | STS12 | STS13 | STS14 | STS15 | STS16 |
|---------|---------|-------|-------|-------|-------|-------|
| | BOW | **0.466/0.496** | 0.376/**0.414** | **0.478/0.482** | 0.424/0.454 | **0.552/0.586** |
| RNN | RNN | 0.323/0.357 | 0.320/0.319 | 0.345/0.345 | 0.402/0.409 | 0.373/0.408 |
| | RNN-concat | 0.419/0.445 | **0.426/0.414** | 0.466/0.452 | **0.497/0.503** | 0.511/0.529 |
| | BOW | 0.497/0.517 | **0.526/0.520** | **0.576**/0.561 | 0.604/0.605 | **0.592/0.592** |
| BOW | RNN | 0.508/0.526 | 0.483/0.489 | 0.575/**0.562** | **0.644/0.641** | 0.585/0.585 |
| | RNN-concat | **0.521/0.540** | 0.491/0.498 | 0.561/0.554 | 0.627/0.625 | 0.584/0.581 |

A corollary is that longer sentences on average have shorter norms, since they contain more words which, in turn, have appeared in more contexts (Adi et al., 2017). During training, the corpus can induce differences in norms in a way that strongly penalises sentences potentially containing multiple contexts, and consequently will disfavour these sentences as similar to other sentences under the dot product. This induces a noise that potentially renders the dot product a less useful metric to choose for STS tasks than cosine similarity, which is unaffected by this issue.

Table 5: Performance of different architectures and sentence representations on unsupervised similarity tasks using dot product as the similarity measure. On each task, the highest performing setup for each encoder type is highlighted in bold and the highest performing setup overall is underlined. All reported values indicate Pearson/Spearman correlation coefficients for the task.

| Encoder | Decoder | STS12 | STS13 | STS14 | STS15 | STS16 |
|---------|---------|-------|-------|-------|-------|-------|
| | BOW | 0.286/**0.384** | 0.381/0.426 | 0.365/0.406 | 0.262/0.392 | 0.260/0.328 |
| RNN | RNN | 0.267/0.286 | 0.371/0.365 | 0.357/0.348 | 0.379/0.379 | 0.313/0.303 |
| | RNN-mean | 0.330/0.361 | **0.420/0.427** | **0.438/0.428** | **0.419/0.426** | 0.324/0.342 |
| | RNN-concat | **0.335**/0.359 | 0.411/0.415 | 0.413/0.403 | 0.414/0.419 | **0.326/0.347** |
| | BOW | 0.351/0.390 | 0.418/0.478 | 0.442/0.488 | 0.455/0.535 | 0.370/**0.461** |
| BOW | RNN | 0.310/0.342 | 0.365/0.422 | 0.396/0.440 | 0.412/0.469 | 0.281/0.330 |
| | RNN-mean | 0.394/0.414 | 0.469/0.495 | 0.490/0.498 | 0.495/0.530 | 0.381/0.439 |
| | RNN-concat | **0.422**/**0.438** | **0.478**/**0.498** | **0.498**/**0.506** | **0.512**/**0.544** | **0.402**/0.460 |

Table 6: Performance of different architectures and sentence representations on supervised transfer tasks. On each task, the highest performing setup for each encoder type is highlighted in bold and the highest performing setup overall is underlined. All reported values indicate test accuracy on the task, except for SICK-R where we report the Pearson correlation with human-provided scores.

| Encoder | Decoder | MR | CR | MPQA | SUBJ | SST | TREC | MRPC | SICK-R | SICK-E |
|---------|---------|-----|-----|------|------|-----|------|------|--------|--------|
| | BOW | 75.78 | 79.34 | 86.25 | 90.77 | 81.99 | 84.60 | 70.55 | 0.80 | 78.81 |
| RNN | RNN | **77.06** | 81.77 | **88.59** | **92.56** | **82.65** | 86.60 | 71.94 | 0.83 | **81.10** |
| | RNN-mean | 76.55 | 81.03 | 87.35 | 92.29 | 81.11 | 84.80 | **73.51** | **0.84** | 78.22 |
| | RNN-concat | 76.20 | **82.07** | 85.96 | 91.80 | 80.83 | **87.20** | 71.59 | 0.82 | 80.35 |
| | BOW | 76.16 | 81.14 | 87.03 | 92.77 | 81.66 | 84.20 | 71.07 | 0.84 | 80.58 |
| BOW | RNN | 76.05 | **82.07** | 85.80 | 92.13 | 80.83 | 87.20 | 72.99 | 0.82 | 78.87 |
| | RNN-mean | 75.85 | 81.30 | 85.54 | 90.80 | 80.12 | 84.00 | 71.13 | 0.81 | 77.76 |
| | RNN-concat | **77.27** | 82.04 | **88.74** | **92.88** | **81.82** | **89.60** | **73.68** | **0.85** | **82.26** |

## C    UNROLLING RNN DECODERS BY TAKING THE MEAN

A practical downside of the unrolling procedure described in Section 3.3 is that concatenating hidden states of the decoder leads to very high dimensional vectors, which might be undesirable due to memory or other practical constraints. An alternative is to instead average the hidden states, which also corresponds to a representation space in which the training objective optimises the dot product as a measure of similarity between a sentence and its context. We refer to this model choice as *-RNN-mean.

Results on similarity and transfer tasks for BOW-RNN-mean and RNN-RNN-mean are presented in Tables 5 and 6 respectively, with results for the other models from Section 5 included for completeness. While the strong performance of RNN-RNN-mean relative to RNN-RNN is consistent with our theory, exploring why it is able to outperform RNN-concat experimentally on STS tasks is left to future work.

