# OpenReview forum: "Decoding Decoders: Finding Optimal Representation Spaces for Unsupervised Similarity Tasks"
_ICLR.cc/2018/Conference — Invite to Workshop Track_

### Official Review · AnonReviewer3 · 2017-11-27
**Gap between theory and results**

**Rating:** 5
**Confidence:** 5

**Review:**

The authors provide some theoretical justification for why simple log-linear decoders perform better than RNN decoders for various unsupervised sentence similarity tasks. They also provide a simple method for improving the performance of RNN based models. Please find below my comments/questions/suggestions:

1) I found the theory to be a bit superficial and there is a clearly a gap between what is proven theoretically and demonstrated empirically. For example, as per the theoretical arguments presented in the paper, RNN-concat should do better than RNN-mean. However, the experiments suggest that RNN-mean always does better and in some cases significantly better (referring to Table 1). How does this empirical observation reconcile with the theory ?

2) The authors mention that the results for SkipThought represented in their paper are lower than those presented in the original SkipThought paper. They say that they elaborate on this in Appendix C but there isn't much information provided. In particular, it would be good to mention the original numbers also (of course I can check the original paper but if the authors provide those numbers then it would ensure that there is no misunderstanding)

3) As mentioned in the previous point, the original SkipThought decoder seems to do better than the modified decoder used by the authors. It is not clear, how this can be justified under the theoretical framework presented by the author. I agree that this could be because in the original formulation (referring to equations in Appendix C) the encoder contributes more directly to the decoder. However, it is not clear how this causes it to be "closer" to the optimal space.  Can this be proven ?

4) Can you elaborate a bit more on how the model is used at test time? Consider the STS17 benchmark and the following sentence pair from it?

- The bird is bathing in the sink.
- Birdie is washing itself in the water basin.

How will you use RNN-mean and RNN-concat to find the similarity between these two sentences.

5) In continuation of the above question, how does the computation of similarity differ when you unroll for 3 steps v/s when you unroll for 5 steps

6) Based on the answers to (4) and (5) above I would like to seek further clarifications on Figure 1.

---

> ### Author Response · Authors · 2017-12-13
> **Thank you**
>
> We would like to thank you for your time and a detailed assessment of our work.
> We hope to address some of your questions and concerns below.
>
>
> > I found the theory to be a bit superficial.
>
> This has been pointed out by all 3 reviewers. We will make our statements more formal and tone down the “theory” aspect in the updated manuscript.
>
>
> > RNN-concat should do better than RNN-mean (but oftentimes the opposite happens)
>
> We have re-run our experiments with dot product instead of cosine similarity as pointed out by Reviewer 2. We found that RNN-concat works much better than previously reported. We will update the manuscript to reflect the changes.
>
>
> > it would be good to mention the original SkipThought numbers
>
> We agree and will do so in the updated manuscript.
>
>
> > The original SkipThought decoder seems to do better than the modified decoder used by the authors
>
> This was definitely a source of concern for us as well. We hypothesise this is because the encoder contributes more directly to the decoder but we cannot say for sure. It is entirely possible the discrepancy is due to variations in experimental setups. E.g. to the best of our knowledge the original paper uses a combination of bidirectional and unidirectional encoders; we use only the latter.
> In this work we did not aim to compete with existing models, we were only interested in testing our assumptions in a fair setting. To this end, we built our experiments upon a well-known TensorFlow Skip-Thought implementation and will make our codebase public.
>
>
> > Can you elaborate a bit more on how the model is used at test time?
>
> Absolutely. The encoder RNN encodes a sentence into some vector f and the decoder RNNs produce sequences of hidden states h(1), h(2), …, h(n) for the previous sentence and g(1), g(2), … g(n) for the next sentence. We simply average or concatenate all those hidden states to form the final sentence representation.
> Importantly, since there are no adjacent sentences during test time, the decoder input for the current step is just the softmax of the previous step multiplied by the word embedding matrix, i.e. w(n) = W*p(n-1).
> Please let us know if that answers your question. We are always happy to elaborate further.
>
>
> > how does the computation of similarity differ when you unroll for 3 steps v/s when you unroll for 5 steps
>
> We would unroll N hidden states of each decoder, so in total we have 2*N vectors of dimension d.
> In RNN-mean, we just average all of them and get 1 vector of dimension d.
> In RNN-concat, we concatenate all of them and get 1 vector of dimension 2*N*d.
> In either case, we use dot product to compare the resulting representations.
>
>
> > I would like to seek further clarifications on Figure 1.
>
> This illustrates the performance on the STS14 task as a function of N, where N is the number of unrolled hidden states of the decoder.
>
>
> Finally, we will notify you when all of the above are addressed in the new version.
>
> Best wishes,
>
> ICLR 2018 Conference Paper816 Authors

---

> ### Author Response · Authors · 2018-01-25
> **Issues addressed in revised version**
>
> Dear Reviewer,
>
> Again thank you for your the time you took to initially assess our work.
>
> We found the comments you made informative, and hope that the newer manuscript manages to address the issues you had with the paper. The reviewer that has currently re-reviewed the manuscript has concluded that the paper is much improved, and has increased their score accordingly. We would very much appreciate it if you could take a look at the updated version with respect to your issues with the initial version.
>
> Specifically:
>
> The theory section is completely rewritten to put it on a more formal, and less superficial ground.
>
> We have now evaluated the models in the dot product space (which is the distance measure we should use according to our analysis), giving expanded experimental results, and find that concat consistently outperforms mean. We discuss the concat-mean situation in an Appendix.
>
> We have added an appendix specifically comparing our models to the original SkipThought implementations for reasonable comparison.
>
> On further investigation, our initial hypothesis regarding the additional conditioning of the decoders at each time step on the encoder output step turned out not to be true, true as the LayerNorm models available for download in fact don't contain this conditioning mentioned in the paper. We have now attributed this difference to differences in experimental setup.
>
> A new diagram demonstrating how to use the unrolling procedure in practice for STS (and other) tasks has been added.
>
> Many thanks.

---

### Official Review · AnonReviewer2 · 2017-11-27
**some contributions, but more concerns; updated paper is an improvement**

**Rating:** 6
**Confidence:** 4

**Review:**

------ updates to review: ---------

I think the paper is much improved. It is much more clear and the experiments are more focused and more closely connected to the earlier content in the paper. Thanks to the authors for trying to address all of my concerns.

I now better understand in what sense the representation space is optimal.  I had been thinking (or perhaps "hoping" is a better word) that the term "optimal" implied maximal in terms of some quantifiable measure, but it's more of an empirical "optimality".  This makes the paper an empirical paper based on a reasonable and sensible intuition, rather than a theoretical result.  This was a little disappointing to me, but I do still think the paper is marginally above the acceptance threshold and have increased my score accordingly.

------ original review below: --------
This paper is about rethinking how to use encoder-decoder architectures for representation learning when the training objective contains a similarity between the decoder output and the encoding of something else. For example, for the skip-thought RNN encoder-decoder that encodes a sentence and decodes neighboring sentences: rather than use the final encoder hidden state as the representation of the sentence, the paper uses some function of the decoder, since the training objective is to maximize each dot product between a decoder hidden state and the embedding of a context word. If dot product (or cosine similarity) is going to be used as the similarity function for the representation, then it makes more sense, the paper argues, to use the decoder hidden state(s) as the representation of the input sentence. The paper considers both averaging and concatenating hidden states. One difficulty here is that the neighboring sentences are typically not available in downstream tasks, so the paper runs the decoder to produce a predicted sentence one word-at-a-time, using the predicted words as inputs to the decoder RNNs. Then those decoder RNN hidden states are used via averaging or concatenation as the representation of a sentence in downstream tasks.

This paper is a source of contributions, but I think in its current form it is not yet ready for publication.

Pros:

I think it makes sense to pay attention to the training objective when deciding how to use the model for downstream tasks.

I like the empirical investigation of combining RNN and BOW encoders and decoders.

The experimental results show that a single encoder-decoder model can be trained and then two different functions of it can be used at test time for different kinds of tasks (RNN-RNN for supervised transfer and RNN-RNN-mean for unsupervised transfer). I think this is an interesting result.

Cons:

I have several concerns. The first relate to the theoretical arguments and their empirical support.

Regarding the theoretical arguments:

First, the paper discusses the notion of an "optimal representation space" and describes the argument as theoretical, but I don't see much of a theoretical argument here.

As far as I can tell, the paper does not formally define its terms or define in what sense the representation space is "optimal". I can only find heuristic statements like those in the paragraph in Sec 3.2 that begins "These observations...".  What exactly is meant formally by statements like "any model where the decoder is log-linear with respect to the encoder" or "that distance is optimal with respect to the model’s objective"?  It seems like the paper may want to start with formal definitions of an encoder and a decoder, then define what is meant by a "decoder that is log-linear with respect to the encoder", and define what it means for a distance to be optimal with respect to a training objective. That seems necessary in order to provide the foundation to make any theoretical statement about choices for encoders, decoders, and training objectives.  I am still not exactly sure what that theoretical statement might look like, but maybe defining the terms would help the authors get started in heading toward the goal of defining a statement to prove.

Second, the paper's theoretical story seems to diverge almost immediately from the choices used in the model and experimental procedure.

For example, in Sec. 3.2, it is stated that cosine similarity "is the appropriate similarity measure in the case of log-linear decoders."  But the associated footnote (footnote 2) seems to admit a contradiction here by noting that actually the appropriate similarity measure is dot product: "Evidently, the correct measure is actually the dot product."  This is a bit confusing.
It also raises a question: If cosine similarity will be used later for computing similarity, then why not try using cosine similarity in place of dot product in the model?  That is, replace "u_w \cdot h_i" in Eq. (2) with "cos(u_w, h_i)".  If the paper's story is correct (and if I understand the ideas correctly), training with cosine similarity should work better than training with dot product, because the similarity function used during training is more similar to that used in testing.  This seems like a natural experiment to try.  Other natural experiments would be to vary both the similarity function used in the model during training and the similarity function used at test time.  The authors' claims could be validated if the optimal choices always use the same choice for the training and test-time similarity functions.  That is, if Euclidean distance is used during training, then will Euclidean distance be the best choice at test time?

Another example of the divergence lies in the use of the skip-thought decoder on downstream tasks. Since the decoder hidden states depend on neighboring sentences and these are considered to be unavailable at test time, the paper "unrolls" the decoder for several steps by using it to predict words which are then used as inputs on the next time step. To me, this is a potentially very significant difference between training and testing. Since much of the paper is about reconciling training and testing conditions in terms of the representation space and similarity function, this difference feels like a divergence from the theoretical story. It is only briefly mentioned at the end of Sec. 3.3 and then discussed again later in the experiments section. I think this should be described in more detail in Section 3.3 because it is an important note about how the model will be used in practice.

It would be nice to be able to quantify the impact (of unrolling the decoder with predicted words) by, for example, using the decoder on a downstream evaluation dataset that has neighboring sentences in it. Then the actual neighboring sentences can be used as inputs to the decoder when it is unrolled, which would be closer to the training conditions and we could empirically see the difference. Perhaps there is an evaluation dataset with ordered sentences so that the authors could empirically compare using real vs predicted inputs to the decoder on a downstream task?

The above experiments might help to better connect the experiments section with the theoretical arguments.

Other concerns, including more specific points, are below:

Sec. 2:
When describing inferior performance of RNN-based models on unsupervised sentence similarity tasks, the paper states: "While this shortcoming of SkipThought and RNN-based models in general has been pointed out, to the best of our knowledge, it has never been systematically addressed in the literature before."
The authors may want to check Wieting & Gimpel (2017) (and its related work) which investigates the inferiority of LSTMs compared to word averaging for unsupervised sentence similarity tasks. They found that averaging the encoder hidden states can work better than using the final encoder hidden state; the authors may want to try that as well.

Sec. 3.2:
When describing FastSent, the paper includes "Due to the model's simplicity, it is particularly fast to train and evaluate, yet has shown state-of-the-art performance in unsupervised similarity tasks (Hill et al., 2015)."
I don't think it makes much sense to cite the SimLex-999 paper in this context, as that is a word similarity task and that paper does not include any results of FastSent. Maybe the Hill et al (2016) FastSent citation was meant instead? But in that case, I don't think it is quite accurate to make the claim that FastSent is SOTA on unsupervised similarity tasks. In the original FastSent paper (Hill et al., 2016), FastSent is not as good as CPHRASE or "DictRep BOW+embs" on average across the unsupervised sentence similarity evaluations. FastSent is also not as good as sent2vec from Pagliardini et al (2017) or charagram-phrase from Wieting et al. (2016).

Sec. 3.3:
In describing skip-thought, the paper states: "While computationally complex, it is currently the state-of-the-art model for supervised transfer tasks (Hill et al., 2016)."
I don't think it is accurate to state that skip-thought is still state-of-the-art for supervised transfer tasks, in light of recent work (Conneau et al., 2017; Gan et al., 2017).

Sec. 3.3:
When discussing averaging the decoder hidden states, the paper states: "Intuitively, this corresponds to destroying the word order information the decoder has learned." I'm not sure this strong language can be justified here. Is there any evidence to suggest that averaging the decoder hidden states will destroy word order information? The hidden states may be representing word order information in a way that is robust to averaging, i.e., in a way such that the average of the hidden states can still lead to the reconstruction of the word order.

Sec. 4:
What does it mean to use an RNN encoder and a BOW decoder?  This seems to be a strongly-performing setting and competitive with RNN-mean, but I don't know exactly what this means.


Minor things:

Sec. 3.1:
When defining v_w, it would be helpful to make explicit that it's in \mathbb{R}^d.

Sec. 4:
For TREC question type classification, I think the correct citation should be Li & Roth (2002) instead of Vorhees (2002).

Sec. 5:
I think there's a typo in the following sentence: "Our results show that, for example, the raw encoder output for SkipThought (RNN-RNN) achieves strong performance on supervised transfer, whilst its mean decoder output (RNN-mean) achieves strong performance on supervised transfer."  I think "unsupervised" was meant in the latter mention.

References:

Conneau, A., Kiela, D., Schwenk, H., Barrault, L., & Bordes, A. (2017). Supervised Learning of Universal Sentence Representations from Natural Language Inference Data. EMNLP.
Gan, Z., Pu, Y., Henao, R., Li, C., He, X., & Carin, L. (2017). Learning generic sentence representations using convolutional neural networks. EMNLP.
Li, X., & Roth, D. (2002). Learning question classifiers. COLING.
Pagliardini, M., Gupta, P., & Jaggi, M. (2017). Unsupervised Learning of Sentence Embeddings using Compositional n-Gram Features. arXiv preprint arXiv:1703.02507.
Wieting, J., Bansal, M., Gimpel, K., & Livescu, K. (2016). Charagram: Embedding words and sentences via character n-grams. EMNLP.
Wieting, J., & Gimpel, K. (2017). Revisiting Recurrent Networks for Paraphrastic Sentence Embeddings. ACL.

---

> ### Author Response · Authors · 2017-12-13
> **Thank you**
>
> Thank you very much for the time and consideration you have taken with your review. We sincerely appreciate your detailed feedback and would like to address some of your questions and concerns below.
>
> > I don't see much of a theoretical argument here.
>
> The lack of formalism was mentioned by all 3 reviewers. We will reformulate this section, give some clearer definitions as you suggested, and tone down the “theory” aspect in general.
>
>
> > … theoretical story seems to diverge almost immediately… if cosine similarity will be used later, then why not try using cosine similarity in place of dot product in the model?
>
> You are absolutely correct. While cosine similarity is clearly related to dot product, it is not a drop-in replacement because it is not always consistent with the dot product. Thank you for pointing out this error in our analysis.
> We will re-run all our experiments with the dot product and will publish the results in the updated manuscript.
> You are also right that in principle one can use cosine similarity, Euclidean distance or indeed any chosen measure in the model. Due to time and computational restrictions, we are unable to run these additional experiments by the rebuttal deadline but we feel these ideas are definitely worth exploring.
>
>
> > if Euclidean distance is used during training, then will Euclidean distance be the best choice at test time?
>
> Our analysis suggests that trying Euclidean distance is a sensible thing to do in this case. Of course, the downstream task might differ so much that the distance and the model itself are not useful at all.
>
>
> > Another example of the divergence lies in the use of the skip-thought decoder on downstream tasks. Since the decoder hidden states depend on neighboring sentences and these are considered to be unavailable at test time, the paper "unrolls" the decoder for several steps by using it to predict words which are then used as inputs on the next time step. To me, this is a potentially very significant difference between training and testing. Since much of the paper is about reconciling training and testing conditions in terms of the representation space and similarity function, this difference feels like a divergence from the theoretical story.
>
> Unfortunately, when the test sentences have no context, the adjacent sentences need to be approximated (e.g. by using the softmax word embeddings or beam search). In those cases the optimal representation is not really attainable but can be “approximated”. However, in other models such as RNN-RNN autoencoders the optimal space is perfectly attainable.
> We feel our work is more about paying attention to the objective as you mentioned. If the model maximises some similarity between representations, it is sensible to try that similarity on the downstream tasks.
>
>
> > It would be nice to be able to quantify the impact (of unrolling the decoder with predicted words) by, for example, using the decoder on a downstream evaluation dataset that has neighboring sentences in it.
>
> We absolutely agree and would love to do this but are unaware of such tasks / datasets. Would the Reviewer be able to help us by suggesting one?
>
>
> > ...The hidden states may be representing word order information in a way that is robust to averaging …
>
> This is a really good point and we totally agree. We will soften or retract our statement.
>
>
> > What does it mean to use an RNN encoder and a BOW decoder? This seems to be a strongly-performing setting and competitive with RNN-mean, but I don't know exactly what this means.
>
> We use the RNN to encode a sentence into a vector h, and then use the bag-of-words decoder, i.e. softmax(U*h), where U is the logit matrix. Please do let us know if you would like a more detailed description of this or any other model.
>
>
> We agree with all other points not mentioned here and will fix accordingly.
>
>
> Finally, we will notify you when all of the above are addressed in the new version.
>
> Best wishes,
>
> ICLR 2018 Conference Paper816 Authors

---

### Official Review · AnonReviewer1 · 2017-11-27
**There are some interesting findings, but not good enough**

**Rating:** 4
**Confidence:** 4

**Review:**

This paper proposes the concept of optimal representation space and suggests that a model should be evaluated in its optimal representation space to get good performance. It could be a good idea if this paper could suggest some ways to find the optimal representation space in general, instead of just showing two cases. It is disappointing, because this paper is named as "finding optimal representation spaces ...".

In addition, one of the contributions claimed in this paper is about introducing the "formalism" of an optimal representation space. However, I didn't see any formal definition of this concept or theoretical justification.

About FastSent or any other log-linear model, the reason that dot product (or cosine similarity) is a good metric is because the model is trained to optimize the dot product, as shown in equation 5 --- I think this simple fact is missed in this paper.

The experimental results are not convincing, because I didn't find any consistent pattern that shows the performance is getting better once we evaluated the model in its optimal representation space.

There are statements in this paper that I didn't agree with

1) Distributional hypothesis from Harris (1954) is about words not sentences.
2) Not sure the following line makes sense: "However, these unsupervised tasks are more interesting from a general AI point of view, as they test whether the machine truly understands the human notion of similarity, without being explicitly told what is similar"

---

> ### Author Response · Authors · 2017-12-13
> **Thank you**
>
> Thank you very much for your time and assessment of our work.
> We hope to address some of your concerns below.
>
> > It could be a good idea if this paper could suggest some ways to find the optimal representation space in general, instead of just showing two cases.
>
> We agree and will describe the general procedure in the updated manuscript.
>
>
> > I didn't see any formal definition of this concept ...
>
> The lack of formalism was mentioned by all 3 reviewers. We will give some clearer definitions and tone down the “theory” aspect in general.
>
>
> > … the reason that dot product is a good metric is because the log-linear model is trained to optimize the dot product
>
> This is exactly what we were trying to say. We also argue that dot product is not necessarily appropriate for comparing RNN encoder vectors. Intuitively, due to non-linearities, small changes in the encoder hidden state might lead to big changes in the decoder outputs and vice versa. We are sorry if our core idea was not clear enough, we will present our analysis better in the updated manuscript.
>
>
> > The experimental results are not convincing
>
> Could you perhaps help us by pointing towards the source of your concerns?
> As our analysis indicates, RNN encoder - RNN decoder is the worst model for similarity because dot product is not appropriate for comparing encoder states when RNN decoder is used. However, dot product is appropriate for BOW encoder - BOW decoder and RNN - RNN-concat. Our experiments show significant and consistent improvements over RNN-RNN.
>
>
> > Distributional hypothesis from Harris (1954) is about words not sentences.
>
> We had no intention to say otherwise. By “sentence-level version of the distributional hypothesis Harris (1954)” we meant that one can think of a “sentence-level version” of the original word-level hypothesis due to Harris (1954). We will fix our sloppy wording in the updated manuscript.
>
>
> > Not sure the following line makes sense: "However, these unsupervised tasks are more interesting from a general AI point of view...
>
> We completely agree with you. In fact, humans are being told what is similar all the time. We will rephrase or retract the statement.
>
>
> Finally, we will notify you when all of the above are addressed in the new version.
>
> Best wishes,
>
> ICLR 2018 Conference Paper816 Authors

---

> ### Author Response · Authors · 2018-01-25
> **Issues addressed in revised version**
>
> Dear Reviewer,
>
> Again thank you for your the time you took to initially assess our work.
>
> We found the comments you made informative, and hope that the newer manuscript manages to address the issues you had with the paper. The reviewer that has currently re-reviewed the manuscript has concluded that the paper is much improved, and has increased their score accordingly. We would very much appreciate it if you could take a look at the updated version with respect to your issues with the initial version.
>
> Specifically:
>
> We now introduce the formalism of the optimal representation space (which you pointed out was missing)
>
> We have more clearly highlighted the reason FastSent works so well in the space it is evaluated in by making a stronger connection to the encoder.
>
> We have analysed the model in both the derived and canonical representation space, yielding an expanded set of results that better empirically supports our theoretical analysis.
>
> We have emphasised that it is a sentence-level version of the distributional hypothesis (although the theory section now takes a different, more formal approach in order to arrive at the semantically similar conclusion - the distributional hypothesis is only required for choosing appropriate outputs).
>
> We have removed the line you pointed out that doesn't make sense.
>
> Many thanks.

---

### Author Response · Authors · 2018-01-05
**Paper revision - formalised theory and dot-product results**

Many thanks to the reviewers for their extensive and valuable feedback.

While the original scope remains the same, the paper itself has changed significantly:
- Greatly expanded theory section, including a thorough definition of an optimal representation space and details how such an optimal space can be discovered for a given model.
- Revisited analysis of BOW and RNN decoders to clarify our argumentation.
- As suggested by the reviewers we now report results using dot-product as similarity metric in the main body of the paper and have moved the results using cosine similarity into the appendix.
- Expanded presentation and discussion of the performance of unrolled RNN decoders.
- Results for mean of unrolled decoder states, and a comparison with (variants of) SkipThought have been added to the appendix.
- Upon further investigation, the SkipThought LayerNorm model, whose results we were comparing against does not directly condition its decoder output on the encoder output at every time step as initially thought. We attribute differences in performance to experimental setup and have thus removed related comments.
- Literature is now reviewed in the introduction (including references suggested by the reviewers).
- An additional figure clarifies the proposed unrolling procedure for RNN decoders.
- Minor changes to the text

We believe we have thoroughly addressed the bulk of the issues highlighted by the reviewers. In particular we now provide a solid theoretical framework for optimal representation spaces and how they can be obtained for a given model. Further we provide results that are more consistent with our theoretical argument and have revisited the majority of the text in the paper.

Again we would like to thank the reviewers for the time and care they have put into their reviews, and would like to invite them to reconsider their original ratings.

---

### Decision · Program_Chairs · 2018-01-29
**ICLR 2018 Conference Acceptance Decision**

**Decision:**

Invite to Workshop Track

**Comment:**

this submission has two results; (1) it defines what it means for the optimal representation is, although this is rather uninteresting that it simply says that if the representation from a model is going to be used based on some given metric, the cost function should directly reflect it, and (2) it shows that different choices of encoding and decoding have different implications. as with most of the reviewers, i found these to be a rather weak contribution.